# Microencapsulation of Flaxseed Oil by Lentil Protein Isolate-κ-Carrageenan and -ι-Carrageenan Based Wall Materials through Spray and Freeze Drying

**DOI:** 10.3390/molecules27103195

**Published:** 2022-05-17

**Authors:** Yingxin Wang, Supratim Ghosh, Michael T. Nickerson

**Affiliations:** Department of Food and Bioproduct Sciences, University of Saskatchewan, 51 Campus Dr., Saskatoon, SK S7N 5A8, Canada; lindsay.wang@usask.ca (Y.W.); supratim.ghosh@usask.ca (S.G.)

**Keywords:** plant protein, polysaccharide, emulsification, omega-3 oil, microencapsulation, spray drying, freeze drying

## Abstract

Lentil protein isolate (LPI)-κ-carrageenan (κ-C) and -ι-carrageenan (ι-C) based microcapsules were prepared through spray-drying and freeze-drying to encapsulate flaxseed oil in order to reach final oil levels of 20% and 30%. Characteristics of the corresponding emulsions and their dried microcapsules were determined. For emulsion properties, all LPI-κ-C and LPI-ι-C emulsions remained 100% stable after 48 h, while the LPI emulsions destabilized quickly (*p* < 0.05) after homogenization mainly due to low emulsion viscosity. For spray-dried microcapsules, the highest yield was attributed to LPI-ι-C with 20% oil, followed by LPI-κ-C 20% and LPI-ι-C 30% (*p* < 0.05). Flaxseed oil was oxidized more significantly among the spray-dried capsules compared to untreated oil (*p* < 0.05) due to the effect of heat. Flaxseed oil was more stable in all the freeze-dried capsules and showed significantly lower oil oxidation than the untreated oil after 8 weeks of storage (*p* < 0.05). As for in vitro oil release profile, a higher amount of oil was released for LPI-κ-C powders under simulated gastric fluid (SGF), while more oil was released for LPI-ι-C powders under simulated gastric fluid and simulated intestinal fluid (SGF + SIF) regardless of drying method and oil content. This study enhanced the emulsion stability by applying carrageenan to LPI and showed the potential to make plant-based microcapsules to deliver omega-3 oils.

## 1. Introduction

Flaxseed oil contains a rich source of polyunsaturated fatty acids such as α-linolenic acid (ALA; ~50%), oleic acid (~20%), and linoleic acid (~15%) [1,2]. ALA, an omega-3 fatty acid, provides health benefits such as developing the brain and nervous system in infants, preventing cardiovascular diseases and immune response disorders, and reducing cholesterol levels [3]. ALA also acts as the precursor of eicosapentaenoic acid (EPA) and docosahexaenoic acid (DHA) for humans [4]. However, polyunsaturated fatty acids are highly susceptible to oxidation under the presence of light, heat, or reactive oxygen species leading to oil rancidity, loss of nutritional value and flavor, and texture and color changes [1]. Microencapsulation can be an effective method to enhance the oxidative stability of flaxseed oil.

Encapsulation is a process for the production of microcapsules with sizes ranging from less than 1 µm to more than 1000 µm, where the bioactive core is entrapped in the wall materials to ensure the functionality of the bioactive compounds and to provide their controlled delivery [5,6,7]. Microencapsulation by complex coacervation followed by drying is one of the most widely used encapsulation techniques for entrapping a lipid core [8]. In terms of production, a coacervate-based emulsion containing the core materials is first prepared, followed by drying, typically spray-drying and freeze-drying, to yield the dried powders [6,9]. Coacervation encapsulation is a simple, solvent-free, and low-cost process, which is excellent for use at industrial scale [10]. Complex coacervation occurs involving the electrostatic attraction between biopolymers such as proteins and polysaccharides during an acid pH titration, where soluble complexes are formed at the initially detectable change in turbidity, and formation of insoluble complexes at further acidification with greater protein–polysaccharide interactions [11,12]. Spray-drying involves the atomization of the emulsion into spherical dry powders with embedded oil at elevated temperature [13]. It is a common drying method used in the food industry due to its low cost and easy handling [5]. However, spray-drying requires wall materials to have good solubility and leads to a great deal of energy waste due to heat loss in the drying chamber and final products with lower oxidative stability [5,8]. Freeze-drying is an attractive drying method due to its simple process and easy handling. Freeze drying is normally applied for heat sensitive products and results in volatile compounds with longer retention time [14]. During the process, the emulsion is frozen between −90 °C to −40 °C, and the pressure is reduced by the vacuum pump, which leads to the sublimation of the frozen water [8]. Nevertheless, freeze-drying requires long processing time, higher cost, and high energy consumption. Studies have been conducted investigating microencapsulated powders prepared by coacervation and subsequent drying. Improved encapsulation efficiency of active compounds was observed in microcapsules of encapsulated palm oil from chitosan-xanthan coacervates with spray-drying [15], β-pinene containing microcapsules from milk proteins-carboxymethylcellulose coacervates and freeze-drying [9], and β-carotene containing microcapsules prepared from chitosan-sodium tripolyphosphate coacervates or chitosan-carboxymethyl cellulose coacervates with freeze-drying [16]. Enhanced oxidative stability was found in tuna oil encapsulated microcapsules with whey protein-gum Arabic via both spray- and freeze-drying [17] and canola oil encapsulated microcapsules with lentil protein-sodium alginate [6].

The selection of biopolymers is critical as their characteristics can greatly affect the coacervation, spray-drying, and freeze-drying processes. The use of plant protein ingredients within the food industry is increasing rapidly due to their lower cost, greater environmental sustainability, perceived safety concerns related to consuming animal products, and consumer dietary preferences [18]. Consumption of plant protein also provides potential health benefits for certain chronic diseases, such as reducing the risk of developing metabolic syndrome and certain types of cancers, improving diabetes management, and promoting weight management [19]. Lentil proteins are gaining tremendous interest because of their non-GMO status, low allergenicity, high solubility among plant proteins, and abundance in Canada [20]. The dominant proteins in lentils are globulin-type (50–65%) and albumin-type (10–25%) proteins [21]. Carrageenan is widely applied in foods because of its texture improving property and its ability to produce products with high encapsulation efficiency in control-delivery systems [22,23]. Carrageenan is a linear sulfated galactans with β (1–3) and α (1–4) linkages that are extracted from red seaweeds in the class *Rhodophyceae* [24]. Carrageenan can be classified into κ-, ι-, and λ-carrageenan with 1, 2, and 3 sulfate groups per disaccharide unit, respectively [24]. Maltodextrin (MD), also known as glucose polymers, is an oligosaccharide obtained from the enzymatic (e.g., α-amylase) or acid hydrolysis of starch, followed by drying [25]. Maltodextrin has dextrose equivalent (DE) value less than 20, and a larger DE value means a higher degree of hydrolysis of starch [26]. Maltodextrin is a common secondary wall material for microencapsulation due to excellent functional properties such as high solubility, low viscosity at higher concentration, and low affinity to hydrophobic compounds [27,28].

Studies have been conducted using protein only to microencapsulate omega-3 oil, such as flaxseed oil containing microcapsules from whey protein concentrate and spray-drying [28], flaxseed oil containing microcapsules from LPI and freeze-drying [29], and fish oil containing microcapsules from soy protein isolate and spray-drying [30]. However, the encapsulation efficiency of oil for these systems was always below 65% when the oil level was 20%. Investigations on using insoluble complexes for microencapsulation have also been conducted, such as tuna oil containing microcapsules from whey protein isolate-gum Arabic insoluble complexes and freeze-drying [17] and β-pinene containing microcapsules from milk proteins-carboxymethylcellulose insoluble complexes and freeze-drying [9]. However, our previous work [31] suggested that more stable emulsions were prepared by using soluble complexes rather than insoluble complexes within the systems of LPI-κ-C and LPI-ι-C at all investigated LPI: carrageenan mixing ratios (i.e., 4:1, 8:1, & 12:1) or homogeneous LPI. This was because soluble complexes were smaller in size compared to LPI aggregates and insoluble complexes, and soluble complexes lowered the interfacial tension to a greater extent. In that work, the most stable emulsion was prepared by soluble complexes at 4:1 LPI: carrageenan (both κ-C and ι-C) at pH 6. Therefore, the objective of this work was to look at the possible use of lentil protein isolate-carrageenan based materials to encapsulate edible oil by first incorporating κ-C and ι-C to LPI to form stable emulsions, along with MD to the system as a coating wall material. Subsequent spray-drying and freeze-drying were then applied to produce microencapsulated powders with better qualities such as higher encapsulation efficiency, oxidative stability, and release properties.

## 2. Results & Discussion

### 2.1. Emulsion Characteristics

Initial emulsions were first homogenized by LPI, LPI-κ-C, or LPI-ι-C solutions at pH 6, followed by a second homogenization with MD solution to alter the final viscosity of the emulsions. Each initial emulsion contained 5% or 7.5% flaxseed oil to deliver 20% or 30% of final oil, respectively, in the dried microencapsulated powders. Details of each formula were given in Appendix A. In terms of naming, for instance, the initial LPI emulsion containing 5% oil was prepared to deliver 20% oil, so it was named LPI 20% in all tables and figures. Emulsion characteristics such as emulsion stability (via creaming), droplet charge, emulsion viscosity, and droplet size were determined (Table 1 and Table 2). For emulsion stability, gravitational separation occurred rapidly after emulsification for LPI 20% and LPI 30% emulsions. Emulsions stabilized by LPI-κ-C or LPI-ι-C complexes showed significant increase in stability against creaming, where all emulsions with 20% or 30% final oil remained 100% stable after 48 h (Table 1).

In terms of droplet size, LPI-κ-C 20% and 30% emulsions formed the smallest droplets (*p* < 0.05) with D_3,2_ of 6.8 µm and 7.7 µm, respectively, and D_4,3_ of 19.6 µm and 21.7 µm, respectively (Table 2). The smaller values of D_3,2_ (*p* < 0.05) for both LPI-κ-C emulsions suggested the formation of larger numbers of smaller droplets. D_3,2_ of LPI 20%, LPI 30%, LPI-ι-C 20%, and LPI-ι-C 30% emulsions were all found to range from 10 to 12 µm (*p* > 0.05). D_4,3_ values of LPI 20% and LPI 30% emulsions were ~20 µm, which were similar to those of the LPI-κ-C 20% and 30% emulsions (*p* > 0.05). LPI-ι-C 20% and 30% emulsions showed the largest D_4,3_ among all systems of 28.8 and 25.1 µm, respectively, (*p* < 0.05) suggesting the presence of some large droplets. Since the average droplet sizes of all systems were comparable, other factors such as droplet charge and viscosity are hypothesized to cause the rapid gravitational separation for the LPI 20% and 30% emulsions. Droplet charge of each emulsion is given in Table 1. The strongest electrostatic repulsion was attributed to LPI-κ-C 20% and 30% emulsions with charges of −66 and −64 mV, respectively. Droplet charge for LPI-ι-C 20% and 30% emulsions was −59 and −61 mV, respectively, which was slightly smaller than both LPI-κ-C emulsions (*p* < 0.05) but still provided significant electrostatic repulsive forces. LPI 20% and 30% emulsions generated the least electrostatic repulsion (*p* < 0.05) against droplet aggregation with droplet charges of −40 and −43 mV. However, the electrostatic repulsion was still considered to be moderate [32].

Emulsion viscosity also has an effect on emulsion stability. The consistency coefficient (m) and the flow behavior index (n) for each system is given in Table 1 (see flow curves, Appendix A). LPI 20% and 30% emulsions showed the lowest viscosity (*p* < 0.05) with m-values of 24 and 20 mPa s, respectively, and this low viscosity led to a remarkable high rate of creaming. Incorporating κ-C and ι-C into the systems both led to higher emulsion viscosity, where LPI-ι-C 20% and 30% emulsions showed the greatest emulsion viscosity (*p* < 0.05) of 270 and 277 mPa s, respectively, followed by LPI-κ-C 30% with m-value of 117 mPa s (*p* < 0.05), and then LPI-κ-C 20% with a value of 69 mPa s (*p* < 0.05). At pH 6, LPI formed soluble complexes with κ-C and ι-C, and excessive carrageenan was contributing to the viscosity to reduce creaming effect, as both carrageenan polysaccharides are good thickening reagents and have good water binding properties [22,32]. Within these two carrageenan polysaccharides, ι-C contains two sulfate groups per disaccharide unit and was able to bind with more water molecules to contribute to a higher viscosity in LPI-ι-C 20% and 30% emulsions [24]. As the oil content of the initial emulsions did not vary (i.e., 5% and 7.5% oil), the viscosity of two emulsions within each type of system showed no difference when compared to each other (*p* > 0.05). The larger D_4,3_ values of LPI-ι-C 20% and 30% emulsions were a result of the higher emulsion viscosities. Studies have suggested that suppression of breaking up droplets during homogenization might occur as the viscosity of an emulsion continuous phase increases [32,33,34]. All emulsions were shear thinning, as shown in the flow curves (Appendix A). The highest n-value was attributed to the LPI-κ-C 20% emulsion, followed by LPI-κ-C 20%, LPI 30%, and LPI 20% emulsions. LPI-ι-C 20% and 30% emulsions showed the smallest n-values, corresponding to their larger average droplet size, and in terms of less homogeneous systems. Viscosity can not only affect emulsion stability but is also extremely important in the subsequent spray-drying process to prepare dried powders. Studies suggest that liquid that is fed into the spray dryer generally should not have its viscosity exceed 300 mPa s to prevent the formation of large or elongated droplets and air inclusion in the droplets during atomization, which in turn will cause negative effects on the dried powder’s characteristics [35]. In this study, all emulsion viscosities were below 300 mPa s.

Confocal laser scanning microscopy images were taken for each fresh initial emulsion (Figure 1). For both LPI 20% and 30% emulsions, very large droplets were observed, confirming the rapid destabilization of the emulsions due to droplet coalescence (Figure 1A,B). In contrast, small droplets were seen in LPI-κ-C 20% & 30% and LPI-ι-C 20% & 30% emulsions with some larger sizes at about 20–25 µm and many below 10 µm, corresponding to their average droplet size (Figure 1C–F; Table 2). The protein aggregates in both LPI 20% and 30% emulsions were also larger in size (marked as green; Figure 1A,B) compared to all other LPI-κ-C and LPI-ι-C emulsions (Figure 1C–F), which suggested that the incorporation of carrageenan could suppress the formation of large protein aggregates, corresponding to our previous findings [36,37]. It was also confirmed that the oil droplets were surrounded by LPI, LPI-κ-C, or LPI-ι-C complexes. A representative image of LPI-ι-C 20% emulsion is provided in Appendix A.

### 2.2. Physical Characteristics of Spray-Drying Powders and Freeze-Drying Powders

#### 2.2.1. Yield of Dried Powders

The initial emulsions were either spray-dried or freeze-dried to give fine powders with 20% and 30% oil. As the LPI 20% and 30% emulsions were highly unstable, these two emulsions were not applied to spray-drying. However, they were applied to freeze-drying since emulsions were pre-frozen at −80 °C. All the solids (i.e., powders) were collected from freeze-drying, but only part of the powders were able to be collected during the spray-drying process. The yields of LPI-ι-C 20%, LPI-ι-C 30%, LPI-κ-C 20%, and LPI-κ-C 30% powders from spray drying were 32.8%, 16.8%, 20.5%, and below 5%, respectively. Due to the very low yield of LPI-κ-C 30% powders, no further experiments were conducted on this system. Yield of powders may have also been affected by oil content in the powders. Both LPI-ι-C and LPI-κ-C powders with 20% final oil showed higher yield than their corresponding powders with 30% oil, which could be attributed to lack of sufficient coverage of wall materials in the oil [38], leading to more samples sticking on the surface of the cylinder and cyclone during drying. Lower yield caused by higher encapsulated oil was also observed in the encapsulation of fish oil in soybean protein powders [30].

#### 2.2.2. Physical Characteristics of Dried Powders

The physical properties, including moisture content, water activity, color, and wettability, of spray drying (LPI-ι-C 20% & 30%, LPI-κ-C 20%) and freeze drying (LPI 20% & 30%, LPI-ι-C 20% & 30%, LPI-κ-C 20% & 30%) powders are given in Table 3. Moisture contents of all spray drying powders were similar, ranging between 2.6% to 3.1% (*p* > 0.05). The freeze-drying powders had moisture contents ranging from 2.7% to 4.1%. It has been suggested by the food industry that dry powders should contain 3–4% moisture to ensure good stability during storage [39,40]. Similar moisture contents were also reported in the encapsulated flaxseed oil-modified starch spray-drying powders [41], kenaf seed oil-sodium caseinate spray-drying powders [42], and flaxseed oil-LPI-MD freeze-drying powders [29]. Water activity (*a_w_*), which measured the availability of free water in foods, was 0.09 for both LPI-ι-C 20% and 30% powders (*p* > 0.05). The *a_w_* of LPI-κ-C 20% powder was significantly lower with *a_w_* value of 0.06 (*p* < 0.05), which might be due to the smaller number of sulfate groups per disaccharide unit to bind less water molecules and lower water solubility of κ-C [43], thereby resulting in a more prominent dehydration during spray-drying process and consequently a lower *a_w_* value. The *a_w_* values for all freeze-drying powders ranged from 0.09 to 0.12, again with LPI-κ-C 30% powder showing the lowest *a_w_* value of 0.09. Generally, *a_w_* should be below 0.60 for food products to protect against microbial spoilage, and for dry powders *a_w_* should not exceed 0.30 [39,44]. It is also agreed that non-enzymatic browning and enzymatic activities are inhibited in foods with *a_w_* around 0.3 [45,46]. Therefore, powders produced in this study should be able to resist chemical and microbial spoilage during storage. However, the *a_w_* values of our powders were lower when compared with those powders with similar moisture contents that are referenced above, whose *a_w_* values ranged from 0.15 to 0.40 [29,41,42]. The lower *a_w_* values for powders in the current study could be attributed to the large amount of MD (dextrose equivalent [DE]: 9–12) in wall materials, and similar *a_w_* values were also observed in spray-drying powders where MD with the same DE was the major wall material [47,48]. The low *a_w_* values might not be ideal to resist lipid oxidation in food products. Some studies suggested that *a_w_* of food products should be near the water monolayer (i.e., *a_w_*: 0.2–0.3) to minimize lipid oxidation [49,50].

Color parameters for each powder including *L* (lightness), *a* (redness), and *b* (yellowness) were given in Table 3. Spray-drying powders showed stronger lightness (*p* < 0.05), with *L* values ranging from 92.4–93.6, than freeze-drying powders, whose *L* values were from 89.2–92.1. Among freeze-drying powders, LPI 20% and 30% powders showed the strongest darkness (*p* < 0.05), which could be attributed to the rapid destabilization of their corresponding emulsions and more oil being separated out. The *a* values of spray-drying powders were all close to zero (*p* > 0.05), while *a* values of freeze-drying powders increased suggesting the stronger redness of freeze-drying powders. Stronger whiteness was observed in spray-drying powders, reflected by the lower *b* values, and freeze-drying powders showed more yellowness due to their higher *b* values. The yellowness increased as the final oil level increased among each wall-type of powders (*p* < 0.05). Wettability of powders, which shows the ability to absorb water, was also reported in Table 3. A shorter time for a product to become reconstituted in water is more desirable [42,51]. Wettability was significantly affected by wall material, and LPI 20% and 30% freeze-drying powders showed significantly stronger wettability than other freeze-drying powders (*p* < 0.05). Addition of polysaccharide to systems leading to lower wettability was also observed in LPI-MD based and LPI-MD-alginate based powders [6]. Wettability was also influenced by drying methods, where all spray-drying powders were more difficult to be wetted than freeze-drying powders (*p* < 0.05). The wettability of each spray-drying powder was similar (*p* > 0.05) and was compatible with other spray-drying powders such as modified starch-based powders [41] and gum Arabic-sodium caseinate-based powders [42].

The average droplet size for each reconstituted emulsion was provided in Table 2. For spray-drying powders, the D_3,2_ values of LPI-ι-C 20%, 30%, and LPI-κ-C 20% reconstituted emulsions were 7.3, 7.3, and 4.1 µm, respectively, and the D_4,3_ values were 20.5, 19.1, and 11.3 µm, respectively. All these values were significantly smaller than their corresponding average values in initially fresh emulsions (*p* < 0.05), attributing to the sample loss or sample degradation during the spray-drying process. These reduction of average droplet size values in the reconstituted emulsions were also observed in the chickpea protein-based powders with 10% flaxseed oil [47]. In terms of freeze-drying reconstituted emulsions, all the D_3,2_ values were similar to their corresponding average values in initial emulsions (*p* > 0.05). However, increased D_4,3_ values (*p* < 0.05) for LPI 20%, 30%, LPI-ι-C 30%, and LPI-κ-C 20% reconstituted emulsions were determined, suggesting the presence of larger droplets, which could be due to coalescence that occurred within the non-encapsulated oils. A layer of flaxseed oil was also observed in both LPI 20% and 30% reconstituted freeze-drying emulsions (data not shown), confirming the destabilization of the emulsions prior to freezing.

### 2.3. Surface Oil and Encapsulation Efficiency of Dried Powders

Surface oil and encapsulation efficiency for each powder were determined and are given in Table 3. The more desirable powders should contain less amounts of surface oil as this negatively affects powder quality such as oxidative stability [47]. Ideally, oil encapsulated powders should have less than 2% surface oil and 98% entrapment efficiency [52]. For spray-drying powders, LPI-ι-C 20% and LPI-κ-C 20% powders showed similar surface oil content at ~3% (*p* > 0.05), and the surface oil increased to 5% for LPI-ι-C 30%, indicating that there was lack of sufficient wall materials (i.e., LPI-ι-C complexes) to encapsulate the oil. An increase in final oil content resulting in an increase in surface oil content was also reported in chickpea protein-based powders [47] and corn zein based powders [53]. The encapsulation efficiency of LPI-ι-C 20%, 30%, and LPI-κ-C 20% powders were 84.1%, 83.5%, and 85.2%, respectively (Table 3). For freeze-drying powders, the lowest surface oil content (*p* < 0.05) was attributed to LPI-ι-C 20% and LPI-κ-C 20% powders with surface oil of 5% and 4.8%, respectively. As the final oil content increased to 30%, the surface oil increased to 13.5% and 14.5% for LPI-ι-C 30% and LPI-κ-C 30% powders, respectively, again indicating the insufficient coverage of oil. The surface oil was very high at 16.2% and 25.2% for LPI 20% and 30% powders, respectively, due to the low stability of their corresponding emulsions. The emulsions started to destabilize prior to freezing at −80 °C, and thus the encapsulation efficiency for LPI 20% and 30% powders was both below 20%. Encapsulation efficiency for LPI-ι-C 20% & 30% and LPI-κ-C 20% & 30% powders was 74.8%, 55.0%, 75.8%, and 51.6%, respectively. It was also observed that surface oil was lower for spray-dried powders than freeze-dried powders (*p* < 0.05). Similar trends were found by Eratte et al. [17], who compared the surface tuna oil contents with whey protein-gum Arabic based spray-drying and freeze-drying powders. Quispe-Condori et al. [53] also reported a lower surface flax oil in corn zein spray-drying powders than in the corresponding freeze-drying powders. One reason might be the loss of some surface oil in the cyclone separator during the spray drying process, resulting in the lower surface oil values of spray-drying powders. Another hypothesis could be attributed to the structural difference between spray-drying and freeze-drying powders (see Section 2.4).

### 2.4. Surface Morphology of Dried Powders

Surface morphology of each spray- and freeze-dried powder was provided in Figure 2 and Appendix A. All the spray-drying powders exhibited wrinkled spherical structure (Figure 2A,B and Appendix A). The sizes of all the spray-drying powders were ranging from below 10 to 30 µm, corresponding to their average droplet sizes. Collapse of some spray-dried LPI-ι-C 30% powders were observed (Figure 2B), again indicating the lack of sufficient wall materials (i.e., LPI-ι-C 30% complexes) to encapsulate the flaxseed oil as oil content increased. In contrast, irregular structure was imaged for all the freeze-drying powders (Figure 2C & D, Appendix A). Some pores were seen in freeze-dried LPI-ι-C 20% powders (Figure 2C), and more pores were found as the oil content of the powders increased to 30% (Figure 2D). The formation of the porous structure in the freeze-dried powders was due to the sublimation of the ice crystals during the primary drying cycle in freeze drying [54]. A similar finding was also reported by Eratte et al. [17], where spray dried powders gave a more compact and wrinkled spherical structure and freeze-drying powders had a more irregular shape and were highly porous. The authors also stated that the porous structure of freeze-dried powders might lead to more leakage of oil and consequently a higher surface oil content. Due to this oil leakage, the higher surface oil content of freeze-drying powders compared to their corresponding spray-drying powders was observed in this study (Table 3).

### 2.5. Lipid Oxidation of Dried Powders

Flaxseed oil is highly susceptible to lipid oxidation due to its high unsaturation level of over 95% [55]. Lipid oxidation can occur chemically with the presence of oxygen through a free radical mechanism to produce primary oxidative products (i.e., peroxides and hydroperoxides), which are highly unstable and will further decompose to form secondary oxidative products (i.e., aldehydes, hydroxy acids, and ketones) [30,56]. Primary lipid oxidation can be determined by measuring peroxide values (PV), and 2-Thiobarbituric acid reactive substances (TBARS) assay can be used to measure secondary lipid oxidation to assess the formation of one of the main resulted compounds—malondialdehyde (MDA) [6]. PV and TBARS values for each powder under 8 weeks of storage are given in Figure 3.

PV value of untreated flaxseed oil before the encapsulation process was 2.1 meq active oxygen/kg oil (Figure 3A). After emulsification and spray-drying processes, the PV values for LPI-ι-C 20%, 30%, and LPI-κ-C 20% powders increased to 7.1, 6.7, and 7.4 meq active oxygen/kg oil, respectively (Figure 3A). The increase in PV values for freeze-drying powders was less prominent (*p* < 0.05) with PV values of 4.3, 3.2, 4.2, and 3.2 meq active oxygen/kg oil for LPI-ι-C 20% & 30% and LPI-κ-C 20% & 30% powders, respectively (Figure 3B). The higher PV values for all the fresh powders compared to untreated oil could be attributed to the emulsification process, where flaxseed oil was blending with solutions and oxygen [47]. In addition to emulsification, heat was also applied in the spray-drying process (inlet and outlet temperature: 135 °C & 95 °C) and was hypothesized to initiate the oxidation more prominently, resulting in a larger increase of PV values in all spray-drying powders. Dried powders with higher PV value compared to untreated oil were also reported in the fish oil-soybean protein-based spray-drying powders [30]. During 8-week of storage, PV values for untreated oil increased two-fold to 4.2 meq active oxygen/kg oil (*p* < 0.05) after 2 weeks, followed by a slow increase to 5.2 meq active oxygen/kg oil at week 4. The increment then became more significantly to 10.0 (*p* < 0.05) and 15.4 (*p* < 0.05) meq active oxygen/kg oil at week 6 and 8, respectively. The smaller increment in the first month could be attributed to less free radicals being present. As there was no protection of the untreated oil during storage, the oil started to react with oxygen to produce peroxides, and the formation of peroxides became more prominent after week 4. Unlike untreated oil, the PV values for all freeze-drying powders were quite stable during 8 weeks of storage (Figure 3A2), where PV values of LPI-ι-C 20% and LPI-κ-C 20% powders showed no difference (*p* > 0.05) in 8 weeks, suggesting the efficiency of the encapsulation process to prevent lipid oxidation. The PV values for LPI-ι-C 30% and LPI-κ-C 30% powders increased slightly (*p* < 0.05) by about 1 meq active oxygen/kg oil from time 0 to week 8, because higher surface oil contents within these two powders were determined. That higher surface oil content led to stronger oil oxidation was also reported in flaxseed oil-chickpea-based powders [47]. For spray-drying powders (Figure 3A1), the increment of PV values for all powders was higher compared to the freeze-drying powders since heat was applied during the spray-drying process. PV values increased slowly for LPI-ι-C 20% powders from 7.1 meq active oxygen/kg oil at time 0 to 11.7 meq active oxygen/kg oil at week 6 (*p* > 0.05), then more significantly to 14.6 meq active oxygen/kg oil at week 8 (*p* < 0.05). Similarly, PV values of LPI-ι-C 30% powders increased gradually from time 0 to week 4 and then more significantly up to week 8. Although the PV values for all spray-drying powders were similar (*p* > 0.05) at time 0, the increment of increase in PV for LPI-κ-C 20% powders during storage was higher, with PV reaching 37.4 meq active oxygen/kg oil at week 8. The surface oil contents were similar in all spray-drying powders, indicating that other factors rather than surface oil contributed a more important role in oil oxidation during storage. One assumption might be the lower *a_w_* value of 0.06 of LPI-κ-C 20% spray-drying powders compared to *a_w_* values of 0.09 for both LPI-ι-C 20% and 30% powders (Table 3). As mentioned above, *a_w_* near the water monolayer (between 0.2–0.3) would show the lowest lipid oxidation. At this point, water molecules could provide a barrier by hydrogen-bonding to lipid hydroperoxides, preventing the direct exposure to air and increasing the relative stability [49,57]. Since the *a_w_* value was very low for LPI-κ-C 20% powder, there was lack of coverage of water against lipid oxidation.

For TBARS, the amount of MDA for untreated oil at time 0 was 0.34 mg/g oil (Figure 3B). Higher level of MDA was determined in freeze-drying powders at time 0 since emulsification was implemented (Figure 3B2). The amount of MDA found in LPI-ι-C 20% & 30% and LPI-κ-C 20% and 30% powders was 0.49, 0.52, 0.45, and 0.64 mg/g oil, respectively. Both powders with 30% oil showed higher MDA values compared to their corresponding powders with 20% oil, which could be attributed to the higher surface oil level. Increment of MDA was more significant (*p* < 0.05) for spray-drying powders at time 0 as both emulsification and heat treatment were applied (Figure 3B1). During 8 weeks of storage, MDA values of untreated oil increased slowly from time 0 to week 3 (*p* > 0.05), then more rapidly starting from week 4 (*p* < 0.05). All freeze-drying powders showed similar growth trends, where the increment of MDA was more steadily from time 0 to week 4 and became more significant (*p* < 0.05) at week 6 and week 8. The increase of MDA value for each freeze-drying powder was about two-fold from time 0 to week 8. The amount of MDA detected for all freeze-drying powders were smaller than MDA measured at week 8 for untreated oil, suggesting the encapsulation process could lower lipid oxidation. For spray-drying powders, the growth of MDA levels was higher than the freeze-drying powders during 8-week of storage, where the increase of MDA of LPI-κ-C 20% powders was the most prominent. The increase of MDA values for LPI-ι-C 20% & 30% spray-drying powders was steadier (*p* > 0.05) from time 0 to week 2 and became more prominent (*p* < 0.05) starting from week 3. For LPI-κ-C 20% powders, MDA values remained stable till week 3 and then increased significantly since week 4. After 8 weeks of storage, MDA value increased two-fold for LPI-ι-C 20% powders, and the increment of MDA was three-fold for LPI-ι-C 30% powders and LPI-κ-C 20% powders.

### 2.6. Oil Release Characteristics of Dried Powders

The flaxseed oil release profile for each powder under simulated gastric fluid (SGF) and sequential exposure to simulated gastric fluid and simulated intestinal fluid (SGF + SIF) is given in Figure 4. In this figure, each result is reported as oil release due to digestive enzyme, which is percentage (%) of oil released from the dried powders under the use of digestive enzymes minus % of oil released from the dried powders without the use of enzymes. Under SGF, the pepsin in the SGF could break down the proteins and led to partial oil being released from the powders. The amount of oil release due to enzyme under SGF for LPI-ι-C 30% and LPI-κ-C 20% spray-drying powders was about 31% (*p* > 0.05), and slightly lower for LPI-ι-C 20% at 26% (*p* < 0.05) (Figure 4A). For freeze-drying powders, the highest amount of oil release under SGF was 37% for LPI-κ-C 20% powders, followed by 34% and 33% for LPI-ι-C 30% and LPI-κ-C 30% powders, respectively. Again LPI-ι-C 20% powders had the least amount of oil released at 26% (*p* < 0.05). This result might suggest that the LPI-ι-C 20% powders were more resistant to the acidic conditions in SGF despite the drying method. Under SGF + SIF, pancreatin broke down the wall materials (i.e., MD and lentil protein) more excessively as pancreatin contains amylase, lipase, and protease. Generally, it was observed that less oil was released (*p* < 0.05) from LPI-κ-C powders than LPI-ι-C powders under SGF + SIF despite the drying method. The amount of oil release for both LPI-ι-C 20% and 30% spray-dried powders was about 56%, followed by LPI-κ-C 20% powders at about 40% (Figure 4A). For freeze-drying powders, over 50% of oil was released from for both LPI-ι-C 20% and 30% powders (*p*> 0.05), followed by LPI-κ-C 20% and 30% powders at about 30% (Figure 4B). The reason for the less oil released in all the LPI-κ-C powders (including both oil contents and both drying methods) was due to the higher amount of oil released without the use of enzymes (data not shown), indicating less resistance of the LPI-κ-C powders against pH change, and the presence of various salts in the SGF during measurement.

## 3. Materials and Methods

### 3.1. Materials

Lentil protein isolate (LPI) used in this study was generously donated by KeyLeaf Corp. (Saskatoon, SK, Canada), and was produced by alkaline extraction (pH 9.5) followed by isoelectric precipitation (pH 4.5), followed by neutralization and spray drying. κ-Carrageenan (κ-C; Lot#: 0001432063) and ι-carrageenan (ι-C; Batch #: 075K1808) were purchased from Sigma-Aldrich Co. (Oakville, ON, Canada). Maltodextrin (MALTRIN M100, dextrose equivalent of 9.0–12.0) was donated by Grain Processing Corporation (Muscatine, IA, USA). Flaxseed oil, which was produced by cold pressing, was provided from Bioriginal Food and Science Crop. (Saskatoon, SK, Canada). All other chemicals used in this study were of reagent grade and purchased through Sigma-Aldrich Co. (Oakville, ON, Canada). Water was filtered through a Milli-Q purification system (Millipore Corporation, Burlington, MA, USA).

### 3.2. Proximate Analysis

Proximate analysis of all materials including moisture, ash, lipid and crude protein (reported on a dry weight basis (d.b.)) was determined according to the Association of Official Analytical Chemists (AOAC) methods 925.10, 923.03, 920.85, and 920.87, respectively [58]. Carbohydrate content was calculated based on percent differential from 100%. Lentil protein isolate (LPI) contained 80.6% (d.b.) protein, whereas κ- and ι-carrageenan had 77.6% (d.b.) and 71.2% (d.b.) carbohydrate, respectively. Maltodextrin contained 99.3% (d.b.) carbohydrate. Complete proximate composition results are given in Appendix A.

### 3.3. Emulsion Characteristics

#### 3.3.1. Emulsion Preparation

Homogeneous LPI, 4:1 LPI-κ-C, and 4:1 LPI-ι-C plus maltodextrin were used as wall materials to encapsulate flaxseed oil. Four hundred milli-litres of each initial emulsion were prepared, and the ratio of oil: (LPI/LPI-κ-C/LPI-ι-C): MD were 5:1:19 and 7.5:1:16.5 (*w*/*w*/*w*) to achieve the final oil level of 20% or 30%, respectively, in dried powders. The details of the compositions of formulations are given in Appendix A. In brief, LPI, κ-C, and ι-C powders were dispersed in Milli-Q water under magnetic stirring at 500 rpm for 16 h stirring at room temperature (21–23 °C). The LPI, LPI-κ-C and LPI-ι-C solutions were then mixed at pH 6 for another hour. The resulting solutions were then mixed with 20 g or 30 g of flaxseed oil at 500 rpm for 15 min. Homogenization process was then applied to the resulting mixture at 15,000 rpm using a rotor stator system (Polytron PT2100 homogenizer) (Kinematica AG, Lucerne, Switzerland) equipped with a 12 mm PT-DA 2112/2EC generating probe for 5 min. In parallel, maltodextrin was dispersed in Milli-Q water for 1 h with magnetic stirring at 500 rpm, and the pH of solution was adjusted to 6. The emulsions were then mixed with maltodextrin solution for another 15 min stirring. The resulting emulsions were then re-homogenized with the same homogenizer at the same speed for another 5 min. All emulsions were prepared in triplicate and reported as the mean ± one standard deviation (*n* = 3).

#### 3.3.2. Emulsion Stability (ES)

ES was determined according to Wang, et al. [36] based on creaming overtime at room temperature (21–23 °C). Ten milliliters of the fresh emulsion was transferred to a 10 mL graduated cylinder (inner diameter = 9.60 mm; height = 114.96 mm; measured by a digital caliper) and allowed to separate for 48 h. The percentage of ES was determined using Equation (1), where V_B_ and V_A_ are the volume of the aqueous phase before emulsification (8.0 mL) and after drainage at each time point, respectively.
(1)$$\%\mathrm{ES}=\frac{V_{B}-{\;V}_{A}}{V_{B}}\times 100\%$$

All measurements were performed in triplicate and reported as the mean ± one standard deviation (*n* = 3).

#### 3.3.3. Droplet Size and Distribution

Each freshly prepared emulsion and reconstituted emulsion (1 g powder in 4 mL Milli-Q water) was placed in a Mastersizer 2000 laser light scattering instrument (Malvern Instruments Ltd., Worcestershire, UK) equipped with a Hydro 2000S sample handling unit at room temperature (21–23 °C) as described by Can Karaca et al. [47] to measure the average droplet size and distribution. The relative refractive index of emulsion was 1.112, which was calculated as the ratio of the refractive index of flaxseed oil (1.479) to the refractive index of water (1.33). The particle size was reported as surface-average diameter (D_3,2_) and volume-average diameter (D_4,3_) expressed using Equations (2) and (3), respectively.
(2)$$D_{3,2}=\frac{\sum_{i=1}N_{i}d_{i}^{3}}{\sum_{i=1}N_{i}d_{i}^{2}}$$
(3)$$D_{4,3}=\frac{\sum_{i=1}N_{i}d_{i}^{4}}{\sum_{i=1}N_{i}d_{i}^{3}}$$
where N_i_ is the total number of particles of diameter (d_i_) [32]. All measurements were performed in triplicate, and two measurements were made for each replicate. The average value of two measurements in each replicate was collected and reported as the mean ± one standard deviation (*n* = 3).

#### 3.3.4. Droplet Charge

Droplet charge of each emulsion was determined using a Zetasizer Nano-ZS90 (Malvern Instruments, Westborough, MA, USA). Ten drops of each of the fresh emulsion were dispersed into 100 mL Milli-Q water at the corresponding pHs, and the dispersions were shaken gently to become homogeneous and used to measure droplet charge.

The electrophoretic mobility (i.e., velocity of a particle within an electric field, U_E_) was used to calculate the zeta potential (ς) by using the Henry equation.
(4)$$U_{E}=\frac{2\varepsilon \times \xi \times f\left(\kappa \alpha \right)}{3\eta }$$
where η is the dispersion viscosity (water in this study), ε is the permittivity, and f(κα) is a function related to the Debye length (κ) and the ratio of particle radius (α) [59]. f(κα) equals 1.5 according to the Smoluchowski approximation. All the measurements were performed at room temperature (22.5 °C) in triplicate, and two measurements were made for each replicate. The average value of two measurements in each replicate was collected and reported as the mean ± one standard deviation (*n* = 3).

#### 3.3.5. Emulsion Viscosity

Each solution or emulsion (~0.62 mL) was placed onto the AR-G2 Rheometer (TA Instruments, New Castle, DE, USA) equipped with a 40 mm diameter 2° acrylic cone [60]. Apparent viscosity was measured as a function of shear rate (2 to 200 s^−1^), and 10 data points were collected per logarithmic decade. Data were then fitted with the power-law model:
log viscosity = (n − 1) log shear rate + log m(5)
where m is the consistency coefficient (equivalent to the apparent viscosity at 1 s^−1^) and n is the flow behavior index. All measurements were performed in triplicate, and the m, n values were reported as the mean ± one standard deviation (*n* = 3).

#### 3.3.6. Confocal Laser Scanning Microscopy (CLSM)

LPI-CAR stabilized emulsions prepared as previously described were imaged through CLSM using 543 and 633 nm lasers [36]. Nile red (0.01% (*w*/*w*)) with a maximum excitation and emission wavelength of 543 nm and 573–613 nm, respectively, was added to the flaxseed oil, with stirring overnight at room temperature (21–23 °C) prior to emulsion preparation. Then 0.1% (*w*/*w*) of fast green in water with the excitation and emission at 633 and 650 nm, respectively, was mixed with emulsions to bind with protein in the continuous phase to reach the final fast green concentration of 0.01% (*w*/*w*). Emulsions were made in triplicate, and four images were taken per slide with the use of a 60× objective lens.

### 3.4. Characteristics of Microencapsulated Powders

#### 3.4.1. Microencapsulation by Spray Drying and Freeze Drying

The freshly prepared emulsions were spray dried using a benchtop Buchi Advanced Mini Spray Drier B-290 (Buchi Labortechnik AG, Flawil, Switzerland) equipped with an atomizing nozzle (0.7 mm diameter). The air flow was set at 473 L/h and 0.41, and the aspiration rate was 38 m^3^/h. The emulsion was stirred at 200 rpm before going into the chamber to avoid droplets aggregation, and the emulsion was pumped at 3 mL/min. The inlet temperature was 135 °C ± 1 °C, and the outlet temperature was 95 °C ± 2 °C. Microcapsules produced from spray drying for each formula were performed in triplicate.

For freeze drying, each emulsion was poured into a small aluminum pan and frozen at –80 °C. All the samples were then freeze dried in a Labconco Free Zone 6 freeze dryer (Labconco Corp., Kansas City, MO, USA). The freeze dryer was initially at –20 °C for a few hours and was then raised to –4 °C for the rest of the drying time until the samples were completely dried.

#### 3.4.2. Recovered Solid Yield

The yield of recovered solid was calculated as the ratio of the powders collected after the drying process and weight of core material and wall materials, shown in the equation below:(6)$$\mathrm{Recovered}\;\mathrm{solid}\;\mathrm{yield}\;\left(\%\right)=\frac{\mathrm{weight}\;\mathrm{of}\;\mathrm{powders}\;\mathrm{collected}\;\left(g\right)}{\mathrm{weight}\;\mathrm{of}\;\mathrm{core}\;\mathrm{and}\;\mathrm{wall}\;\mathrm{materials}\;\left(g\right)}\times 100\%$$

#### 3.4.3. Physical Characteristics

Moisture content of microcapsules was determined according to the Association of Official Analytical Chemists (AOAC) methods 925.10 moisture [58]. The water activity (*a_w_*) of microcapsules was measured with an AquaLab 4TE water activity meter (Decagon Devices, Inc., Pullman, WA, USA) with a 0.001 sensitivity at 22 °C.

The color of microcapsules was measured using a Hunter Colorimeter (ColorFlex EZ 45/0, Hunter Associates Laboratory, Inc., Reston, VA, USA), and the *L* (lightness), *a* (redness), and *b* (yellowness) were reported.

Wettability of microcapsules was measured according to Chew, et al. [42] with some modification. 0.5 g of microcapsules was added into 50 mL Milli-Q water with magnetic stirring at 150 rpm at room temperature (21–23 °C). The time of microcapsules to reach full dissolution was recorded.

All the measurements were performed in triplicate, and two measurements were made for each replicate. The average value of two measurements in each replicate was collected and reported as the mean ± one standard deviation (*n* = 3).

#### 3.4.4. Surface Oil and Encapsulation Efficiency

Surface oil and encapsulation efficiency were determined based on Haq and Chun [61] with some modifications. A 50 mL centrifuge tube was used to weigh 1 g of microcapsules and 30 mL of hexane. The tube was then vortexed for 30 s to extract the surface oil. The solvent was then filtered twice using #1 Whatman filter paper (Whatman International Ltd., Maidstone, UK), and the organic solvent was collected in a 100 mL beaker and evaporated in a fume hood overnight, followed by drying at 105 °C for 30 min. The surface oil of microcapsules was then weighed. Surface oil (SO) and entrapment efficiency (EE) were calculated based on the definition:(7)$$\mathrm{SO}=\frac{\mathrm{weight}\;\mathrm{of}\;\mathrm{surface}\;\mathrm{oil}\;\mathrm{on}\;\mathrm{the}\;\mathrm{microcapsules}\;\left(g\right)}{\mathrm{weight}\;\mathrm{of}\;\mathrm{microcapsules}\;\left(g\right)}\times 100\%$$
(8)$$\mathrm{EE}=\frac{\mathrm{total}\;\mathrm{oil}\;\left(\%\right)-\mathrm{surface}\;\mathrm{oil}\;\left(\%\right)}{\mathrm{total}\;\mathrm{oil}\;\left(\%\right)}\times 100\%$$
where total oil is the oil content of 20% or 30%. Measurements were performed in triplicate and reported as the mean ± one standard deviation (*n* = 3).

#### 3.4.5. Microcapsule Morphology

Surface morphology of the spray- and freeze-dried powders was imaged using a scanning electron microscope (SEM) of Phenom G2Pure (Phenom-World, Eindhoven, The Netherlands). The microcapsules were coated with gold and imaged at 1000× magnification.

#### 3.4.6. Oxidative Stability

Oxidative stability of the encapsulated flaxseed oil in the microcapsules was determined by measuring peroxide value (PV) and 2-thiobarbituric acid reactive substances (TBARS) every week of storage over an 8-week period at room temperature (21–23 °C). Modified from Can Karaca, et al. [47], encapsulated oils were extracted by dissolving about 1.5 g microcapsules in 6 mL Milli-Q water, followed by 15 min-stirring at 500 rpm. A mixture of 40 mL hexane/isopropanol (3:1, *v*/*v*) was added to extract the oil with another 15 min stirring. Aluminum foil was applied to the beaker to prevent light-induced oil oxidation. The resulting mixtures were centrifuged at 4193× *g* for 15 min at room temperature. The organic solvent (top layer) was then transferred to a 250 mL Erlenmeyer flask and dried under a stream of nitrogen in the fume hood. The oxidative stability of the untreated flaxseed oil was used as the control sample, and the same extraction process was applied. Only the capsule powders using mixed wall materials were examined since the LPI-20% and 30% formed poor capsules due to emulsion instability.

##### Peroxide Value (PV)

The peroxide value (PV) experiment was conducted according to Chang, et al. [6] and Koç, et al. [62]. About 0.2 g of flaxseed oil was mixed with 10 mL of acetic acid/chloroform solution (3:2, *v*/*v*) and 200 µL of saturated potassium iodide (KI). The solution was allowed to react for 1 min with occasional shaking, and 10 mL of Milli-Q water was added to stop the reaction. The solution was then titrated with 0.001 N sodium thiosulfate (Na_2_S_2_O_3_) with the presence of 1% (*w*/*v*) cooked corn starch indicator until the violet color disappeared. PV of oil was calculated using the following formula:(9)$$\mathrm{PV}=\frac{\left(S-B\right)\times N\times 1000}{W}$$
where S is the volume (mL) of Na_2_S_2_O_3_ solution used to titrate the encapsulated and untreated oils, B is the volume (mL) of Na_2_S_2_O_3_ solution used to titrate the blank (without oils), N is the normality of Na_2_S_2_O_3_ solution, and W is the oil weight (g). Measurements were performed in triplicate and reported as the mean ± one standard deviation (*n* = 3).

##### 2-Thiobarbituric Acid Reactive Substances (TBARS)

TBARS test was performed according to Akhlaghi and Bandy [63] with some modifications. Firstly, 50 μL of 8.1% (*w*/*v*) SDS, 375 μL of 20% acetic acid (~pH 3.5), 375 μL of 0.8% (*w*/*v*) 2-thiobarbituric acid (TBA), 8.25 μL of 0.02% (*w*/*v*) butylated hydroxytoluene (BHT) (in dimethyl sulfoxide (DMSO)) and 200 μL of the emulsified oil mixture (about 15 mg of oil dissolved in 10 mL of 2% SDS-20 mM acetic acid-sodium acetate buffer at pH 3.5) were added in a 2.0 mL centrifuge tube. The blank was prepared under the same experimental conditions except that 200 μL of 2% SDS-20 mM acetic acid-sodium acetate buffer was added. A standard curve was prepared using malondialdehyde (MDA) (0.25–5 nM) under the same conditions. Samples and standards were then heated at 95 °C for 1 h, followed by 5 min cooling and 10 min centrifugation at 4000× *g* (Eppendorf Centrifuge 5424, Hamburg, Germany). Absorbance at 532 nm for all solutions was measured. TBA values were expressed as mg MDA/g oil. All the measurements were performed in triplicate, and two measurements were made for each replicate. The average value of two measurements in each replicate was collected and reported as the mean ± one standard deviation (*n* = 3).

#### 3.4.7. Oil Release Characteristics

The release profile of the encapsulated flaxseed oil was examined by an in vitro assay simulating gastric and intestinal conditions with some modifications [47]. Simulated gastric fluid (SGF; 1 L at pH 1.2) contained 3.2 g pepsin, 2 g NaCl, and 7.0 mL 36% HCl, and simulated intestinal fluid (SIF; 1 L at pH 6.8) was prepared by adding 10 g pancreatin, 6.8 g K_2_HPO_4_. For exposure to SGF, approximately 1 g of microcapsules was mixed with 10 mL SGF and incubated in a shaking water bath at 100 rpm for 2 h at 37 °C. The solution was then vortexed with the 15 mL of hexane to extract the released oil, followed by centrifugation at 4193× *g* for 10 min at room temperature. The oil was determined gravimetrically similar to previously described. For exposure to SGF and SIF in sequence, 1 g of powder was mixed with the 10 mL SGF and incubated under the same conditions. The pH of the resulting solution was then adjusted to pH 6.8 with the addition of 10 mL SIF, followed by 3 h incubation at 37 °C at 100 rpm. The solution was then vortexed with the 30 mL of hexane to extract the released oil, and the oil was determined gravimetrically. In parallel, the same conditions, except that no digestive enzyme was added, were applied to each sample to measure the amount of oil release without enzymes. Finally, each oil release value was reported as the percentage of oil released under the use of digestive enzymes minus the percentage of oil released without the use of digestive enzymes. Measurements were performed in triplicate and reported as the mean ± one standard deviation (*n* = 3). Only the mixed wall materials were used since the LPI-20% and 30% formed poor capsules due to emulsion instability.

### 3.5. Statistics

A one-way analysis of variance with a Scheffe test was performed to determine differences for all emulsions and reconstituted emulsions in emulsion stability, consistency coefficient and flow behavior index of emulsion viscosity, droplet size (D_3,2_, D_4,3_), and droplet charge. This was also used to determine the difference for powders within each type of drying method for water activity, moisture, color (*L, a, b*), wettability, surface oil, and encapsulation efficiency. Peroxide value and TBARS were determined through a Tukey HSD test. The independent *t*-test was performed to assess the differences of droplet size (D_3,2_, D_4,3_) between the initial emulsion and its corresponding reconstituted spray- or freeze-dried emulsions. All statistics were performed using SPSS software (IBM, Armonk, NY, USA).

## 4. Conclusions

This study prepared microencapsulated powders through complex coacervation of LPI-κ-C and LPI-ι-C by spray-drying and freeze-drying. Initial emulsion stability was enhanced remarkably by using LPI-κ-C and LPI-ι-C complexes compared to homogeneous LPI, which could be mainly attributed to the improvement of emulsion viscosity. Emulsions were then either spray-dried or freeze-dried. All the solids were able to be recovered for freeze-dried powders, but significant loss was found during the spray-drying process, with the highest yield of 33% for LPI-ι-C 20% oil powders. The physical characteristics of each powder including moisture content, *a_w_*, color, wettability, and droplet size were determined. For *a_w_* values, all the powders in this study showed low *a_w_* values ranging from 0.06 to 0.12. Spray-dried powders showed lower surface oil content and higher encapsulation efficiency compared to freeze-dried powders, possibly due to the more compact structure of the spray-dried powders which provided better protection. However, powders produced by spray-drying showed very strong oil oxidation during 8 weeks of storage, which was mainly attributed to the heat effect during the spray-drying process. Encapsulation of flaxseed oil through freeze-drying showed significantly lower oxidation than the untreated oil, suggesting that the encapsulation process was successful. The oil release profile suggested that more oil was released due to digestive enzymes for LPI-κ-C powders under SGF and more oil was released due to digestive enzymes for LPI-ι-C powders under SGF + SIF, regardless of drying method and oil content.

Learning from the results obtained from the current study, more work can be done in the future to improve the quality of the microencapsulated powders. For instance, a high energy emulsification process can be applied to form emulsions with higher stability and to find out if the yield of the spray-drying powders can be improved, as Di Giorgio et al. [30] and de Barros Fernandes et al. [64] reported the increase in yield of powders after applying ultrasonic emulsification. In this study, no powders within both drying methods with surface oil below 2% were achieved. In order to achieve the food industry’s goal, some improvements could be implemented in future studies, such as increasing the concentration of LPI-ι-C and LPI-κ-C to achieve better coverage of oil droplets (e.g., from 1% LPI-ι-C to 2% LPI-ι-C). Moreover, it seems that the lipid oxidation was accelerated by the very low *a_w_* value; therefore, the replacement of part of MD to other simple sugars such as sucrose can be taken into consideration. A study has reported that the incorporation of sucrose to the spray-drying powders could effectively increase *a_w_* of the powders [48].

## Figures and Tables

**Figure 1 molecules-27-03195-f001:**
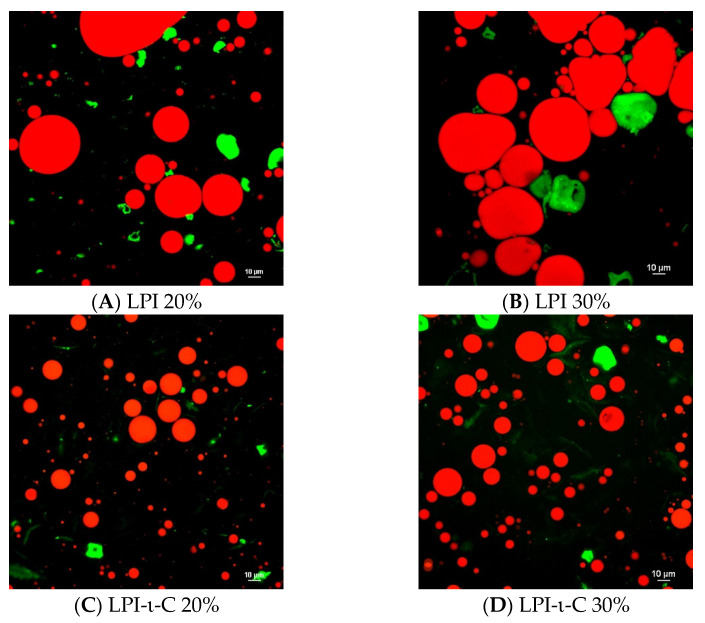
Confocal laser scanning microscopy (CLSM) images of oil-in-water emulsions for systems of LPI 20% and 30% ((**A**,**B**), respectively), LPI-ι-carrageenan (LPI-ι-C) 20% and 30% ((**C**,**D**), respectively), and LPI-κ-carrageenan (LPI-κ-C) 20% and 30% ((**E**,**F**), respectively). Red color represents oil droplets, and green color represents proteins. All the scale bars represent 10 µ. The percentage on the name of the samples represents the final oil content in dried powders.

**Figure 2 molecules-27-03195-f002:**
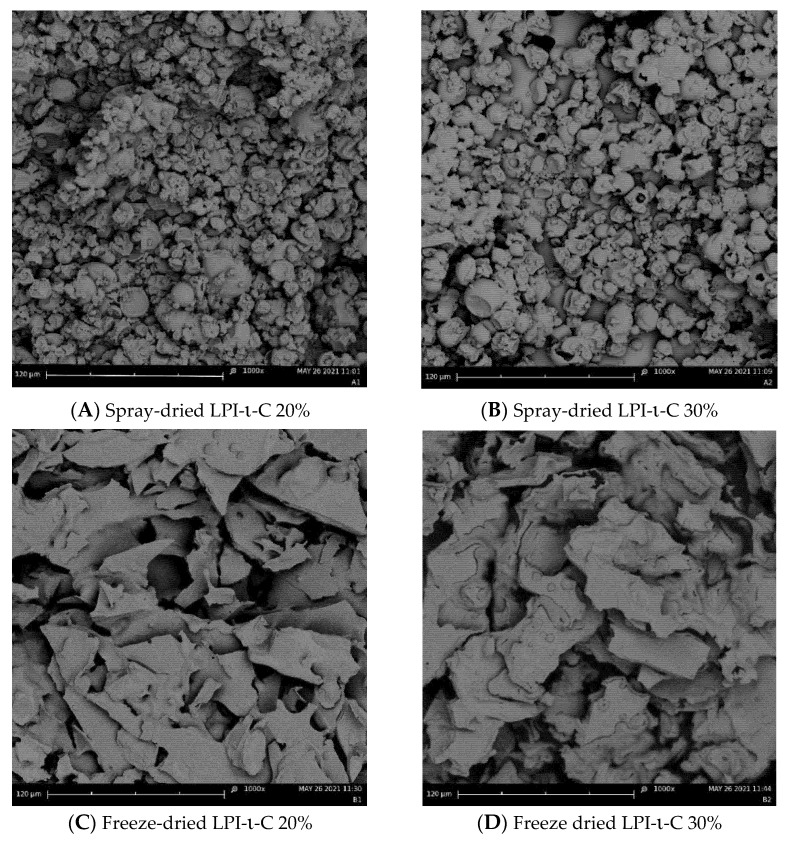
Scanning electron microscopy (SEM) images of the spray-dried LPI-ι-carrageenan (LPI-ι-C) 20% and 30% powders ((**A**,**B**), respectively), and freeze-dried LPI-ι-C 20% and 30% powders ((**C**,**D**), respectively) at 1000× magnification. All the scale bars represent 120 µm.

**Figure 3 molecules-27-03195-f003:**
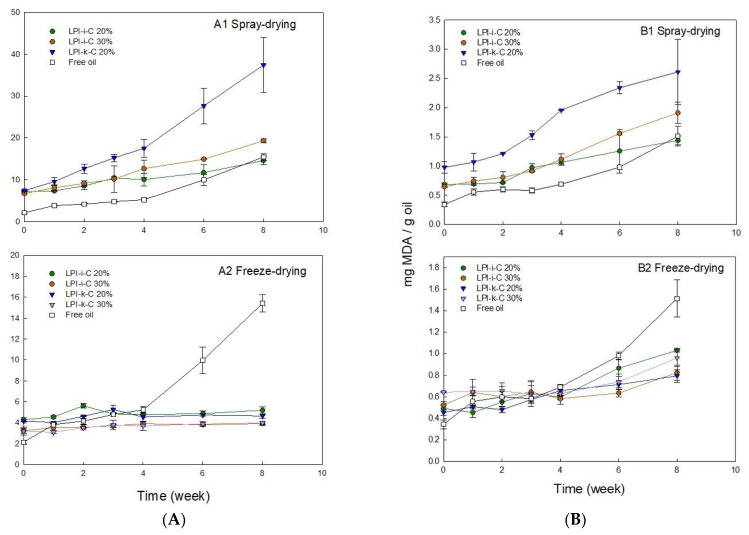
Changes in (**A**) peroxide value (PV) and (**B**) 2-thiobarbituric acid reactive substances (TBARS) for the untreated and encapsulated flaxseed oil through spray-drying (1) and freeze-drying (2) over 8 weeks of storage at room temperature (21–23 °C). Note that untreated oil was not spray dried nor freeze dried. Both LPI 20% & 30% powders were not included in the oxidation tests as their corresponding emulsions rapidly destabilized after emulsification. The percentage on the name of the samples represents the final oil content in dried powders. The *Y*-axis is different between spray- and freeze-drying in order to show the trends. Data represent the mean ± one standard deviation (*n* = 3).

**Figure 4 molecules-27-03195-f004:**
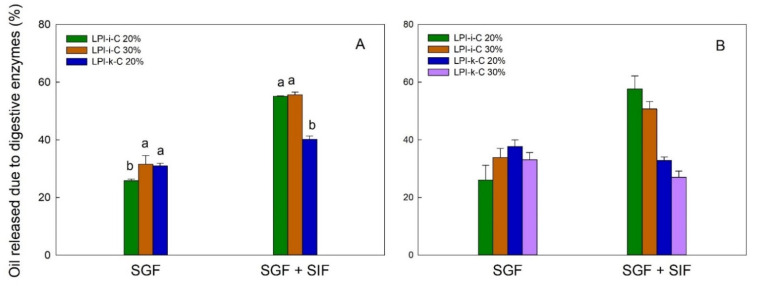
The release of flaxseed oil due to digestive enzymes from spray-drying (**A**) and freeze-drying (**B**) microcapsules under simulated gastric fluid (SGF) and sequential exposure to simulated gastric fluid and simulated intestinal fluid (SGF + SIF). Each value is reported as the percentage of oil released with the use of digestive enzymes minus the percentage of oil released without the use of digestive enzymes. Data represent the mean ± one standard deviation (*n* = 3). The percentage on the name of the samples represents the final oil content in dried powders.

**Table 1 molecules-27-03195-t001:** Emulsions characteristics for emulsions stabilized by homogeneous lentil protein isolate (LPI) solutions and mixtures of LPI with κ-, ι-carrageenan at pH 6 to deliver 20% and 30% oil. Data represents the mean ± one standard deviation (*n* = 3).

Type of Emulsions ^1^	Emulsion Stability at 48 h (%)	Droplet Charge (mV)	Consistency Coefficient (mPa s)	Flow Behavior Index
LPI 20%	n.d.	−40.40 ± 0.95 ^d^	23.7 ± 4.4 ^d^	0.69 ± 0.04 ^bc^
LPI 30%	n.d.	−43.43 ± 1.58 ^d^	19.9 ± 5.4 ^d^	0.74 ± 0.04 ^ab^
LPI-ɩ-C 20%	100.0 ± 0.0 ^a^	−58.80 ± 0.82 ^c^	270.2 ± 7.9 ^a^	0.61 ± 0.01 ^c^
LPI-ɩ-C 30%	100.0 ± 0.0 ^a^	−61.27 ± 1.64 ^bc^	277.3 ± 4.8 ^a^	0.62 ± 0.00 ^c^
LPI-κ-C 20%	100.0 ± 0.0 ^a^	−65.55 ± 1.24 ^a^	68.6 ± 13.2 ^c^	0.83 ± 0.03 ^a^
LPI-κ-C 30%	100.0 ± 0.0 ^a^	−63.92 ± 0.98 ^ab^	116.8 ± 26.3 ^b^	0.75 ± 0.03 ^ab^

Emulsion stability was not determined for LPI 20% and 30% emulsions as they rapidly destabilized a few minutes after emulsification. Droplet charge and Power-law parameters were determined immediately after homogenization. Abbreviation: n.d.: not determined; LPI: lentil protein isolate; ι-C: ι-carrageenan; κ-C: κ-carrageenan. ^1^ Type of emulsions: the percentage on the name of the samples represents the final oil content in dried powders. STAT was carried out to compare the difference for all samples within each property, and different lowercase letters (a–d) indicate significantly different at *p* < 0.05.

**Table 2 molecules-27-03195-t002:** Droplet size of initial emulsions and reconstituted emulsions. Data represents the mean ± one standard deviation (*n* = 3).

Type of Emulsions ^1^	Initial Emulsions		Reconstituted Spray-Drying Emulsions		Reconstituted Freeze-Drying Emulsions	
	Droplet Size D_3,2_ (µm)	Droplet Size D_4,3_ (µm)	Droplet Size D_3,2_ (µm)	Droplet Size D_4,3_ (µm)	Droplet Size D_3,2_ (µm)	Droplet Size D_4,3_ (µm)
LPI 20%	11.85 ± 0.58 ^a^	20.74 ± 0.42 ^bc^	n.d.	n.d.	11.25 ± 1.01 ^a^	23.84 ± 1.24 ^b^
LPI 30%	12.66 ± 1.71 ^a^	21.12 ± 2.95 ^bc^	n.d.	n.d.	11.84 ± 0.35 ^a^	27.13 ± 1.24 ^ab^
LPI-ɩ-C 20%	10.83 ± 0.18 ^a^	28.78 ± 1.82 ^a^	7.33 ± 0.09 ^a^	20.48 ± 1.70 ^a^	10.57 ± 0.79 ^a^	29.80 ± 2.27 ^a^
LPI-ɩ-C 30%	10.34 ± 0.45 ^a^	25.06 ± 0.68 ^ab^	7.30 ± 0.28 ^a^	19.14 ± 1.63 ^a^	10.47 ± 0.11 ^a^	32.02 ± 2.50 ^a^
LPI-κ-C 20%	7.68 ± 0.28 ^b^	19.60 ± 1.27 ^c^	4.05 ± 0.15 ^b^	11.34 ± 0.66 ^b^	8.22 ± 0.40 ^b^	23.29 ± 1.30 ^b^
LPI-κ-C 30%	6.77 ± 0.46 ^b^	21.66 ± 0.79 ^bc^	*	*	7.34 ± 0.58 ^b^	23.80 ± 1.88 ^b^

* indicates the very low yield of the powders of LPI-κ-C 30%. Abbreviation: n.d.: not determined; LPI: lentil protein isolate; ι-C: ι-carrageenan; κ-C: κ-carrageenan. ^1^ Type of emulsions: the percentage on the name of the samples represents the final oil content in dried powders. STAT was carried out to compare the difference for all samples within each average droplet size, and different lowercase letters (a–c) indicate significantly different at *p* < 0.05.

**Table 3 molecules-27-03195-t003:** Physical characteristics of spray-drying and freeze-drying powders. Data represents the mean ± one standard deviation (*n* = 3).

Type of Powders ^1^	Water Activity	Moisture	Color			Wettability (s)	Surface Oil (%)	Encapsulation Efficiency (%)
			*L*	*a*	*b*			
**Spray drying**								
LPI-ɩ-C 20%	0.090 ± 0.008 ^a^	2.57 ± 0.21 ^a^	93.58 ± 0.31 ^a^	0.00 ± 0.06 ^a^	10.35 ± 0.39 ^b^	441 ± 27 ^a^	3.19 ± 0.15 ^b^	84.05 ± 0.75 ^a^
LPI-ɩ-C 30%	0.093 ± 0.005 ^a^	2.96 ± 0.45 ^a^	92.39 ± 0.06 ^b^	−0.01 ± 0.07 ^a^	13.10 ± 0.35 ^a^	459 ± 21 ^a^	4.95 ± 0.65 ^a^	83.51 ± 2.16 ^a^
LPI-κ-C 20%	0.064 ± 0.009 ^b^	3.07 ± 0.29 ^a^	93.44 ± 0.22 ^a^	−0.02 ± 0.03 ^a^	10.58 ± 0.53 ^b^	412 ± 8 ^a^	2.95 ± 0.28 ^b^	85.23 ± 1.38 ^a^
LPI-κ-C 30%	-	-	-	-	-	-	-	-
**Freeze drying**								
LPI 20%	0.107 ± 0.005 ^ab^	3.91 ± 0.30 ^ab^	89.68 ± 0.06 ^c^	0.36 ± 0.02 ^b^	16.41 ± 0.19 ^c^	82 ± 7 ^b^	16.21 ± 0.26 ^b^	18.94 ± 1.30 ^c^
LPI 30%	0.115 ± 0.007 ^a^	3.39 ± 0.15 ^b^	89.21 ± 0.27 ^c^	0.54 ± 0.10 ^a^	19.08 ± 0.18 ^a^	101 ± 11 ^b^	25.17 ± 0.25 ^a^	16.10 ± 0.84 ^c^
LPI-ɩ-C 20%	0.103 ± 0.007 ^ab^	4.08 ± 0.17 ^a^	91.94 ± 0.19 ^ab^	0.12 ± 0.05 ^c^	14.11 ± 0.22 ^d^	239 ± 19 ^a^	5.04 ± 0.37 ^d^	74.82 ± 1.84 ^a^
LPI-ɩ-C 30%	0.115 ± 0.004 ^a^	3.50 ± 0.05 ^b^	91.06 ± 0.12 ^b^	0.31 ± 0.02 ^b^	17.77 ± 0.15 ^b^	271 ± 12 ^a^	13.51 ± 0.74 ^c^	54.95 ± 2.48 ^b^
LPI-κ-C 20%	0.102 ± 0.009 ^ab^	3.88 ± 0.14 ^ab^	92.13 ± 0.07 ^a^	0.08 ± 0.04 ^c^	13.52 ± 0.29 ^d^	234 ± 15 ^a^	4.84 ± 0.47 ^d^	75.80 ± 2.37 ^a^
LPI-κ-C 30%	0.086 ± 0.012 ^b^	2.68 ± 0.05 ^c^	91.56 ± 0.63 ^ab^	0.23 ± 0.02 ^bc^	16.25 ± 0.76 ^c^	235 ± 6 ^a^	14.51 ± 0.66 ^c^	51.64 ± 2.21 ^b^

Notes: LPI (20 and 30%) emulsions were too unstable for the spray-drying process. LPI-κ-C 30% showed too low a yield in the spray dryer for further measurements. Statistics were run within each type of drying method. Surface oil was calculated as weight of surface oil on the powders divided by weight of powders. Encapsulation efficiency was calculated as the difference of total oil and surface oil divided by total oil. Abbreviation: n.d.: not determined; LPI: lentil protein isolate; ι-C: ι-carrageenan; κ-C: κ-carrageenan. ^1^ Type of emulsions: the percentage on the name of the samples represents the final oil content in dried powders. STAT was carried out to compare the difference for all samples within each property, and different lowercase letters (a–d) indicate significantly different at *p* < 0.05.

## Data Availability

Data is contained within the article or Appendix A.

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
