# Peer review of "Microencapsulation of Flaxseed Oil by Lentil Protein Isolate-κ-Carrageenan and -ι-Carrageenan Based Wall Materials through Spray and Freeze Drying"

_molecules, 2022, doi:10.3390/molecules27103195_

Round 1
Reviewer 1 Report
The authors aimed to evaluate different combinations (Table S2) of lentil protein isolate (LPI: 20%, 30%) + carrageenan to encapsulate flaxseed oil, incorporating either κ-carrageenan (C) and i-C to stabilize emulsions, maltodextrin (MD) as an additional coat and spray-drying or freeze-drying methods. As expected, all six experimental treatments before and after drying exhibit specific emulsion (Table 1) and droplet (Table 2) characteristics, physicochemical & morphometric characteristics, and peroxidation & digestibility patterns.
The way in which the authors discuss their results is more descriptive than inductive, yet comparative with the findings of other similar studies. Minor changes could improve the scientific soundness of the study as follows:
- Title. OK
- Abstract. It should be more concise without sacrificing important results, expressed in a more quantitative way, including statistical differences (p-values). If the authors agree to include the statistical analyzes discussed below, include the relevant data.
- Introduction. This section should be shorter, and any argumentative statement should be placed in the discussion section.
- Figures. Their resolution should be improved (≥300 dpi, particularly figure 3) and the B&W color scale could be better. Figure 3 All the graphs included in figure three could be fitted as kinetic curves by finding the equations that fit each behavior (goodness of fit). Kinetic data could reveal more valuable information.
- Tables. Statistical differences between samples are indicated but not referred as footnotes. Please include the meaning of each treatment code as footnote.
- Results and discussion. Be consistent with all abbreviations throughout the manuscript, including their meanings the first time they are mentioned. Even though the discussion is well supported with descriptive data (Tables & figures), the authors should intent to give a deeper explanation as to the associated factors related to inter-sample variations. Additional statistical analyzes using grouping techniques (e.g., PCA, HCA) or multiple linear regression could help to weigh the effect of carrageenan type, its concentration, and the method of drying. It is suggested to check the following studies as an example (DOI): 10.1016 / j.ijbiomac.2021.04.165, 10.1111 / jtxs.12164
Author Response
Abstract: has been revised with p values added.
Introduction: We had conflicting reviewer requests to make it shorter or longer. We feel the length was necessary to introduce all of the relevant topics (flaxseed oil, lentil protein, microencapsulation, drying methods, coacervation, carrageenan, maltodextrin) and give justification for our study and experimental design. We added a few more references. An argumentative statement was written as the objective.
Figures: All the figures have been replaced with the resolution larger or equal 300dpi. After reviewing the data, we feel doing kinetic analysis would not be that informative. There are 7 points total for each line, so the data set would be small. The analysis would not be that significantly powerful to make strong conclusions.
Tables: footnotes have been revised.
Results and discussion: We feel additional statistics are not warranted and would be over reaching. For instance, we only have two drying methods, and we weren’t able to have the same samples for both. E.g., the spray drying once does not have LPI-kappa-car 30%. Sample size is just too small for the different variables. Abbreviations have been revised.
Spelling and grammar were checked throughout the manuscript.
Reviewer 2 Report
Molecules - 1469919
Microencapsulation of flaxseed oil by lentil protein isolate-κ- carrageenan and -ι-carrageenan based wall materials through spray and freeze drying
The work reports the application of carrageenan isolated from lentil protein as wall material for encapsulationg flaxseed oil comparing spray and freeze drying.
The work presents as main novelty the combination of the material encapsulated, and the wall material. The experimental design is appropriate and the results are well discussed. The paper can be accepted with very minor comments, listed below, where I suggest the authors to simply highlight the importance of the combination of materials.
Lines 63-71: I suggest the authors to add further works to support the importance of the current work. The authors may feel free to add any references, but I suggest the following:
Improvement of the functional and antioxidant properties of rice protein by enzymatic hydrolysis for the microencapsulation of linseed oil - https://doi.org/10.1016/j.jfoodeng.2019.109761
Improving the emulsifying property of potato protein by hydrolysis: an application as encapsulating agent with maltodextrin - https://doi.org/10.1016/j.ifset.2021.102696
Microencapsulation of Flaxseed Oil by Spray Drying: Effect of Oil Load and Type of Wall Material - https://doi.org/10.1080/07373937.2012.696227
Influence of emulsion composition and inlet air temperature on the microencapsulation of flaxseed oil by spray drying - https://doi.org/10.1016/j.foodres.2010.10.018
Similar comments to lines 73-77:
IMPROVEMENT OF FUNCTIONAL PROPERTIES OF RAPESEED PROTEIN CONCENTRATES PRODUCED VIA ALCOHOLIC PROCESSES BY THERMAL AND MECHANICAL TREATMENTS - https://doi.org/10.1111/j.1745-4549.2009.00476.x
Role of plant protein in nutrition, wellness, and health - 10.1093/nutrit/nuz028
Author Response
Spelling and grammar were checked throughout the manuscript.
Line 63-71 are now revised as the following:
Nevertheless, freeze-drying requires long processing time, higher cost, and high energy consumption. Studies have been conducted on investigating microencapsulated powders prepared by coacervation and subsequent drying. Improved encapsulation efficiency of active compounds was observed in microcapsules of encapsulated palm oil by chitosan-xanthan coacervates with spray-drying (Rutz, Borges et al. 2017), β-pinene containing microcapsules by milk proteins-carboxymethylcellulose coacervates and freeze-drying (Koupantsis, Pavlidou et al. 2014), and β-carotene containing microcapsules prepared by chitosan-sodium tripolyphosphate coacervates or chitosan-carboxymethyl cellulose coacervates with freeze-drying (Rutz, Borges et al. 2016). Enhanced oxidative stability was found in the tuna oil encapsulated microcapsules with whey protein-gum Arabic with both spray- and freeze-drying (Eratte, Wang et al. 2014) and canola oil encapsulated microcapsules with lentil protein-sodium alginate (Chang, Varankovich et al. 2016).
Line 73-77:
The use of plant protein ingredients within the food industry is increasing rapidly due to their lower cost, greater environmental sustainability, perceived safety concerns related to consuming animal products, and consumer dietary preferences (Campos-Vega, Loarca-Piña et al. 2010). Consumption of plant protein also provides potential health benefits on certain chronic diseases, such as reducing the risk of developing metabolic syndrome and certain types of cancers, improving diabetes management, and promoting weight management (Ahnen, Jonnalagadda et al. 2019). Lentil proteins are gaining tremendous interest because of their non-GMO status, low allergenicity, high solubility among plant proteins, and abundance in Canada (Liang and Tang 2014).